# A green-lipped mussel reduces pain behavior and chondrocyte inflammation and attenuated experimental osteoarthritis progression

JooYeon Jhun[1☯¤a], Hyun Sik Na[1☯¤a], Keun-Hyung Cho[1¤a], Jiyoung Kim[1], Young-Mee Moon[1], Seung Yoon Lee[1¤a], Jeong Su Lee[1¤a], A. Ram Lee[1¤a], Seok Jung Kim[2], Mi-La Cho[1‡¤a¤b]*, Sung-Hwan Park[3‡]*

1 The Rheumatism Research Center, Catholic Research Institute of Medical Science, The Catholic University of Korea, Seoul, South Korea, 2 Department of Orthopedic Surgery, Uijeongbu St. Mary's Hospital, College of Medicine, The Catholic University of Korea, Seoul, Republic of Korea, 3 Division of Rheumatology, Department of Internal Medicine, Seoul St. Mary's Hospital, College of Medicine, The Catholic University of Korea, Seoul, Republic of Korea

☯ These authors contributed equally to this work.
¤a Current address: Department of Biomedicine & Health Sciences, College of Medicine, The Catholic University of Korea, Seocho-gu, Seoul, Republic of Korea
¤b Current address: Department of Medical Lifescience, College of Medicine, The Catholic University of Korea, Seocho-gu, Seoul, Republic of Korea
‡ These authors also contributed equally to this work
* iammila@catholic.ac.kr (M-LC); rapark@catholic.ac.kr (S-HP)

**Data Availability Statement:** All relevant data are within the manuscript.

## Abstract

The green-lipped mussel (GLM) contains novel omega-3 polyunsaturated fatty acids, which exhibit anti-inflammatory and joint-protecting properties. Osteoarthritis (OA) is a degenerative joint disease characterized by a progressive loss of cartilage; oxidative stress plays a role in the pathogenesis of OA. The objectives of this study were to investigate the in vivo effects of the GLM on pain severity and cartilage degeneration using an experimental rat OA model, and to explore the mode of action of GLM. OA was induced in rats by intra-articular injection of monosodium iodoacetate (MIA) into the knee. Oral GLM was initiated on the day after 3dyas of MIA injection. Limb nociception was assessed by measuring the paw withdrawal latency and threshold. Samples were analyzed both macroscopically and histologically. Immunohistochemistry was used to investigate the expression of interleukin-1β (IL-1β), IL-6, nitrotyrosine, and inducible nitric oxide synthase (iNOS) in knee joints. Also, the GLM was applied to OA chondrocyte, and the expression on catabolic marker and necroptosis factor were evaluated by real-time polymerase chain reaction. Administration of the GLM improved pain levels by preventing cartilage damage and inflammation. GLM significantly attenuated the expression levels of mRNAs encoding matrix metalloproteinase-3 (MMP-3), MMP-13, and ADAMTS5 in IL-1β-stimulated human OA chondrocytes. GLM decreased the expression levels of the necroptosis mediators RIPK1, RIPK3, and the mixed lineage kinase domain-like protein (MLKL) in IL-1β-stimulated human OA chondrocytes. Thus, GLM reduced pain and cartilage degeneration in rats with experimentally induced OA. The chondroprotective properties of GLM included suppression of oxidative damage and inhibition of

**Funding:** This research was supported by a grant of the Food Industry Promotional Agency of Korea. The funders had no role in study design, data collection and analysis, decision to publish, or preparation of the manuscript.

**Competing interests:** The authors have declared that no competing interests exist.

**Abbreviations:** GLM, The green-lipped mussel; OA, Osteoarthritis; MIA, monosodium iodoacetate; IL, interleukin; iNOS, inducible nitric oxide synthase; MLKL, mixed lineage kinase domain-like protein; MMP, matrix metalloproteinase; ECM, extracellular matrix; RIP, receptor interacting protein kinase 1; DHA, docosahexaenoic acid; NF-κB, nuclear factor kappa-light-chain-enhancer of activated B cells.

catabolic factors implicated in the pathogenesis of OA cartilage damage. We suggest that GLM may usefully treat human OA.

## Introduction

Osteoarthritis (OA) is the most widespread form of arthritis, which is a dynamic disease of the articular cartilage, the underlying bones, and the synovia of joints. The degenerative disease is characterized by articular cartilage destruction caused by an anabolic/catabolic imbalance imparted by mechanical stress [1,2]. OA features low-grade inflammation. Inflammatory cytokines, chemokines, reactive oxygen species (ROS), and matrix metalloproteinase (MMPs) are released by chondrocytes and related cells in the joints of patients with OA [3–5]. The proinflammatory cytokines and chemokines compromise the homeostasis of the cartilage matrix of OA joints, associated with increased production of interleukin-1β (IL-1β), which induces chondrocytes to produce inflammatory mediators such as nitrotyrosine and IL-6, which further increase the harmful cellular responses [6]. And previously studies have reported the nuclear factor-kappaB (NF-κB) transcription factors as over activated in OA and as a diseases-providing factor [7,8]

Necroptosis (programmed necrosis), a recently recognized form of cell death, triggers serious inflammation [9]. Necroptotic cells feature damage-associated molecular patterns that trigger robust inflammatory responses; necroptosis is markedly more immunogenic than apoptosis, associated with high-level inflammation and extracellular matrix (ECM) degradation [10]. Necroptosis is mediated principally by the receptor interacting protein kinase 1 (RIP1), the receptor interacting protein kinase-3 (RIP3), and the mixed lineage kinase domain-like protein (MLKL) [11,12]. RIP3 expression was increased in patients with OA. Necroptosis inhibitors could serve as OA therapeutics [13].

A green-lipped mussel (GLM) has been prepared from a significant commercial marine species of New Zealand. The GLM is a natural dietary supplement that reduces OA inflammation and arthritis. GLM contains abundant long-chain omega-3 polyunsaturated fatty acids, docosahexaenoic acid (DHA), and many minor fatty acids (including 5, 9, 12, 16-nonadecatetraenoic acid, 5, 9, 12, 15-octadecatetraenoic acid, and 5, 9, 12, 15, 18-heneicosapentaenoic acid) [14]. Many animal and clinical studies have shown that GLM exhibits anti-inflammatory, anti-arthritic, and gastro-protective properties [15–18]. Specifically, long-chain omega-3 polyunsaturated fatty acids such as EPA and DHA are known to exert anti-inflammatory effects. Also, previous study showed that osteoarthritis development is notably associated with oxidative stress and ROS [19,20]. There are many studies of its application in the oral administration of osteoarthritis through anti-inflammatory and antioxidant effects [21,22]. In particular, GLM have antioxidant activity. Bioactive peptides of GLM with the strongest radical scavenging activity and ACE inhibitory activity would supply a therapeutic effect for the osteoarthritis [23].

Given the possible significance of catabolic and necroptotic pathways in OA pathogenesis, and the anti-inflammatory and anti-oxidative effects of GLM, we explored the effects of GLM in an animal model of OA. GLM reduced pain, cartilage destruction, regional catabolism, inflammation, and injury in a rat model of OA. Apart from its chondroprotective effects, GLM inhibited catabolic factors and necroptotic markers involved in OA pathogenesis.

## Materials and methods

### Animals

Seven-week-old male Wistar rats weighing 180~250 g at the start of the experiment were purchased from Shizuoka Laboratory Center (Shizuoka, Japan). A maximum of three

animals per cage were housed in a room featuring controlled temperature (20–26˚C) and light (12-h/12-h light/dark cycle) conditions. The rats had free access to a gamma-ray-sterilized diet (Teklad Global 18% Protein Rodent Diet, Harlan Laboratories, Inc., Indianapolis, IN, USA) and autoclaved water. All animal research procedures were conducted in accordance with the Laboratory Animals Welfare Act (Korea), the Guide for the Care and Use of Laboratory Animals, and the Guidelines and Policies for Rodent Experiments of the Institutional Animal Care and Use Committee (IACUC) of the School of Medicine, the Catholic University of Korea. The IACUC and the Department of Laboratory Animals of the Catholic University of Korea, Songeui Campus, accredited the Korean Excellence Animal Laboratory Facility in co-operation with the Korean Food and Drug Administration in 2017 and full accreditation by AAALAC International followed in 2018. All experimental procedures were evaluated and conducted in accordance with the protocols approved by the Animal Research Ethics Committee at the Catholic University of Korea (Permit Number: CUMC 2020-0244-01). All procedures performed in this study followed the ethical guidelines for animal use.

## MIA-induced OA

The rats were randomly assigned to treatment groups prior to the beginning of the study. Rats were divided into six groups (6 rats in each group) and were group housed in two cages. Each in vivo experiment was repeated a total 2 times. After anesthetization with isoflurane, the rats were injected with 3 mg monosodium iodoacetate (MIA) (Sigma, St. Louis, MO, USA) dissolved in 50 μL (60 mg/mL) volume using a 26.5-G needle inserted through the patellar ligament into the intra-articular space of the right knee. GLM at 100, 300mg/kg and celecoxib at 50 mg/kg, were administered orally to MIA-injected rats.

## Osteoarthritic pain assessment

Nociception in MIA-treated rats was tested using a dynamic plantar aesthesiometer (Ugo Basile, Gemonio, Italy). This is an automated version of the von Frey hair assessment tool and is used to assess mechanical sensitivity. Assessment was conducted by placing each rat on a metal mesh in an acrylic chamber housed in a temperature-controlled room (20–26˚C), where the rat rested for 10 min before the touch stimulator unit was positioned under the animal. An adjustable angled mirror was used to place the stimulating microfilament (0.5 mm in diameter) below the plantar surface of the hind paw. When the instrument was activated, the fine plastic monofilament advanced at a constant speed and touched the paw in the proximal metatarsal region. The filament exerted a gradually increasing force on the plantar surface, beginning below the threshold of detection, and increasing until the stimulus became painful, as indicated by the rat's withdrawal of its paw. The force required to elicit a paw-withdrawal reflex was recorded automatically and measured in g. A maximum force of 50 g and a ramp speed of 25 s were used for all aesthesiometer tests.

## Weight balance assessment

The weight balance of MIA-treated rats was analyzed using a capacitance meter (IITC Life Science, Victory Blvd, CA, USA). Each rat was allowed to acclimate for 5 min in an acrylic holder. Then, both feet were fixed to the pad and the weight balance measured over 5 s. This was repeated twice. The weights the unguided and guided legs were calculated and entered into a formula that yielded a percentage; this compared the legs with and without OA.

## Histopathological analysis

Knee joints were collected 3 weeks after MIA induction. The tissues were fixed in 10% (v/v) formalin for 24 h, decalcified using Decalcifying Solution-Lite (Sigma) for 72 h, and embedded in paraffin. Sections 5-μm in thickness were cut, dewaxed using xylene, dehydrated through alcohol gradient baths, and stained with safranin O.

## Immunohistochemistry

Paraffin-embedded sections were incubated at 4°C with the following primary monoclonal antibodies: Anti-cluster of differentiation (CD)4, anti-CD19, anti-interleukin (IL)-1β, anti-IL-6, anti-inducible nitric oxide synthase (iNOS), anti-nuclear factor kappa-light-chain-enhancer of activated B cells (NF-κB), anti-matrix metalloproteinase (MMP)-1, and anti-MMP3. The samples were then incubated with the appropriate, secondary biotinylated antibodies, followed by a 30-min incubation with a streptavidin-peroxidase complex. The reaction products were developed using the 3, 3-diaminobenzidine chromogen (Dako, Glostrup, Denmark). Immuno-histochemistry evaluations were performed independently by two experienced researchers who were blinded to the study group. The positive cell percentage was analyzed with HDAB (Hematoxylin & DBA) by selecting color deconvolution in the plugin item in the image J program (NIH, MD USA).

## Human chondrocyte isolation and differentiation

Human articular cartilage was obtained from five OA patients treated in Uijeongbu St. Mary's Hospital (UC14CNSI0150) where they underwent knee arthroplasty or joint replacement surgery. All five patients gave fully informed written consent in accordance with the relevant ethical guideline. Chondrocytes were isolated as previously described [24]. Briefly, OA joints were chopped finely and digested with 5 mg/mL protease at 36°C for 1 h and then with 2 mg/mL collagenase and 0.5 mg/mL hyaluronidase (all from Sigma-Aldrich) for 3 h. After several subcultures, chondrocytes below passage 5 were seeded at $5 \times 10^4$ cells/well into 24-well plates. Following 24 h of starvation, the chondrocytes were stimulated with 20 ng/mL IL-1β or various concentrations of GLM (10, 100, 250 μg/mL) in serum-free Dulbecco's Modified Eagle's Medium (DMEM) for 48 h. The supernatants and cell layers were collected for further analyses.

## Enzyme-linked immunosorbent assays (ELISAs)

The concentrations of IL-6 and MCP-1 in cell supernatants were measured using sandwich ELISAs (Duoset; R&D Systems, Minneapolis, MN, USA). After color development, the absorbance was measured at 405 nm on an ELISA microplate reader (Molecular Devices, Sunnyvale, CA, USA).

## Real-time Polymerase Chain Reaction (PCR)

Total RNA was isolated from human chondrocytes with the aid of the TRI reagent (Molecular Research Center, Cincinnati, OH, USA) according to the manufacturer's instructions and reverse-transcribed using a first-strand cDNA synthesis kit (Takara, Shiga, Japan). qPCR was performed with the aid of a LightCycler 2.0 instrument (Roche Diagnostics Indianapolis, IN, USA; software version 4.0) and the SensilFAST SYBR Kit (Bioline, Cincinnati, OH, USA). The housekeeping β-actin gene served as the normalization control. The primer sequences were as follows: β-actin, forward 5′-GGACTTCGAGCAAGAGATGG-3′, reverse 5′-TGTGTTGGGGTACA GGTC TTTG−3′; MMP3 forward: 5′-CTCACAGACCTGACTCGGTT-3′, reverse 5′-CACGCCT GAAGGAAGAGA TG-3′; MMP13 forward: 5′-CTATGGTCCAGGAGATGAAG-3′, reverse

5′-AGAGTCTTGCCTGTATCCTC-3′; MCP1 forward 5′-CAGCCAGATGCAATCAATGC-3′, reverse 5′-GTGGTCCATGGAATCCTGAA-3', MLKL forward 5′-GCAAGCTGGTGGCTGTGA AC-3′, reverse 5′-TTCATCCACAGAGGGCCGCA-3′; and RIPK1 forward 5′-GACGAAGCCAA CTACCATCTT-3′, reverse 5′-TCTCCTTTCCTCCTCTCTGTT-3′; RIPK3 forward 5′-ATGTC GTGCGTCAAGTTATGG-3′, reverse 5′-CGTAGCCCCACTTCCTATGTTG-3′;

## CCK8 assay

Chondrocyte proliferation was measured using the Cell Counting Kit-8 (Dojindo Laboratories, Kumamoto, Japan) as previously described [25]. Chondrocytes were seeded into 96-well plates at $1x10^4$ cells/well. Cells were pre-incubated for 24 h in serum-free DMEM with the Insulin-Transferrin Selenium-A (ITSA) solution (ThermoFisher, Waltham, MA, USA) followed by stimulation with 20 ng/mL IL-1β or GLME (10, 100, 250 μg/mL) for 48 h. Cells were supplemented with the CCK-8 solution for another 3 h at 37˚C and the $OD_{450}$ measured using a microplate reader (Molecular Devices).

## Statistical analysis

Statistical analyses were performed using the nonparametric Mann-Whitney $U$-test for comparisons between two groups, and one-way ANOVA with the Bonferroni post-hoc test for multiple comparisons. GraphPad Prism (ver. 5.01; GraphPad Software Inc., San Diego, CA, USA) was used for all analyses. The data are presented as means ± standard deviations (SDs). For all comparisons, $P < 0.05$ was taken to indicate statistical significance.

## Results

### GLM regulates the pain threshold

The pain-killing capacity of GLM was analyzed; pain was quantified using the electronic von Frey system. Weight bearing was assessed employing a capacitance meter. Three weeks after MIA injection into the right knees, we found that the paw withdrawal latency and withdrawal threshold were dose-dependently reduced by GLM compared to the vehicle (Fig 1A and 1B); GLM also improved the weight balance (Fig 1C).

### GLM suppresses cartilage destruction and inflammation in the rat model of MIA-induced OA

We collected the right knee joints 3 weeks after MIA injection and used safranin O staining for histopathological analysis. GLM dose-dependently reduced cartilage damage and proteoglycan depletion (Fig 2A) as assessed by OARSI and Mankin scoring (Fig 2B). Immunohistochemistry confirmed the presence of inflammatory mediators and the aging factor. Rheumatoid arthritis-like synovitis was observed after OA induction (Figs 2C and 3A). The T- and B-cell immune responses were reduced by GLM at 300 mg/kg compared to the vehicle, as were the levels of IL-1β, IL-6, iNOS, and NF-κB (Figs 2D and 3B).

### GLM regulates the expression of catabolic enzymes and pro-inflammatory cytokines in human OA chondrocytes

We used immunohistochemistry to explore the levels of catabolic factors. We previously found that GLM inhibited cartilage destruction. We confirmed that GLM regulated the levels of catabolic factors (Fig 4A). The MMP-1 and MMP-3 levels were reduced by treatment with GLM (300 mg/kg) (Fig 4B). We explored whether GLM regulated catabolic gene expression in OA

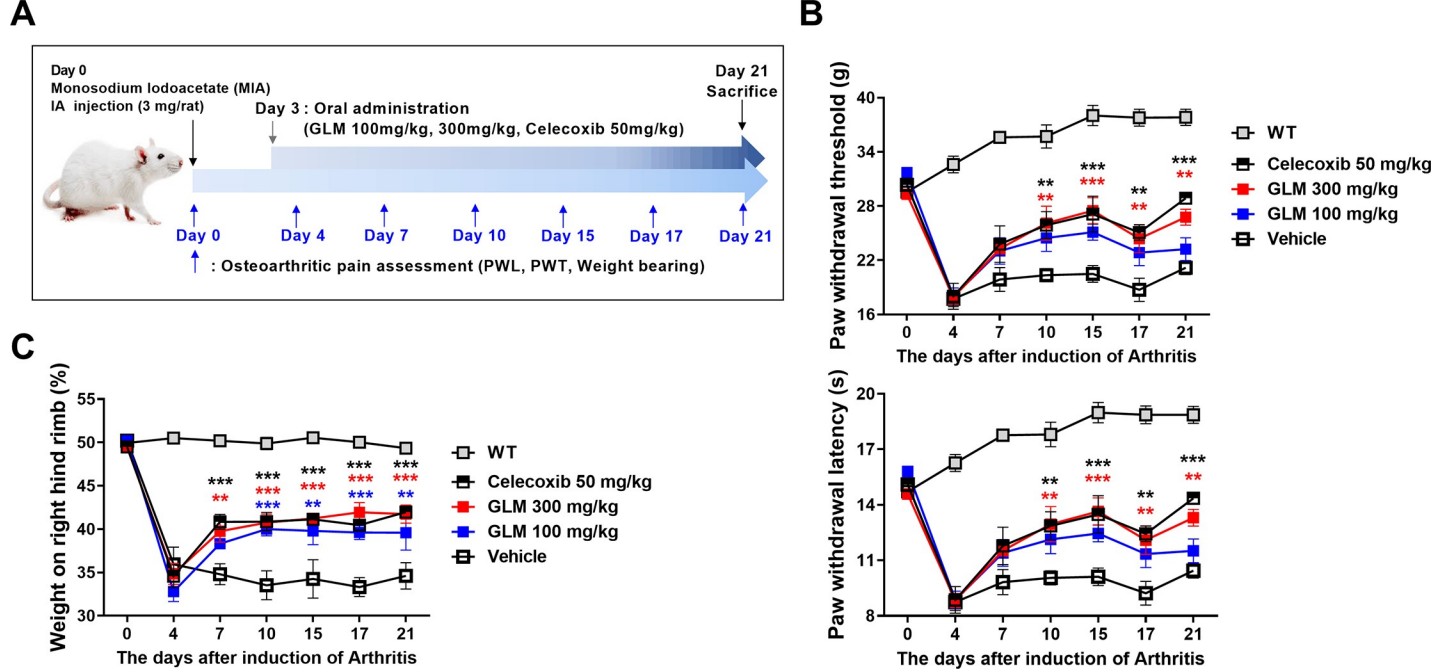

**Fig 1. Green-lipped mussel extract reduced pain in MIA-injected rats.** (A) Schedule of rat experimental procedure is shown by a diagram. (B) Pain was analyzed as the difference between the PWL (left) and PWT (right) in vehicle-treated MIA-induced OA rats, and those given GLME orally. Rats with MIA-induced OA (n = 6 per group) were evaluated to day 21. Nociceptive testing was performed using a dynamic plantar esthesiometer; this is an automated version of the von Frey hair assessment tool. (C) Weight-bearing was examined (n = 6 per group) today 28. Data are presented as the means ± SDs of those of three independent experiments. Significant differences were evident between the vehicle- and GLME-treated groups. *P < 0.05, **P < 0.01, and ***P < 0.001 compared to the vehicle-treated group.

chondrocytes. We found that the levels of mRNAs encoding MMP3 and MMP13 (ECM degradation proteins of articular cartilage) were significantly downregulated in the GLME compared to the control group (Fig 4C). In addition, the level of MCP-1, which chemoattracts macrophages, was also decreased by GLM treatment (Fig 4C). The concentrations of IL-6 and MCP-1 in culture supernatants were also reduced by GLM (Fig 4D). To control for possible GLM cytotoxicity, we performed the CCK-8 assay. GLM (10 μg/mL, 100 μg/mL) did not affect the proliferation of OA chondrocytes (Fig 4E).

## GLM inhibits inflammatory cell death of human OA chondrocytes

We next tested whether GLM modulated necroptosis of human OA chondrocytes, as was true of the cells in MIA-induced OA cartilage tissue. We immunohistochemically stained for RIP1, RIP3, and pMLKL (markers of necroptosis) to determine whether the anti-nociceptive effect of GLM reflected an anti-necroptosis activity, perhaps explaining the protective effect of GLM in terms of structural damage to knee joints. Synovial RIP1, RIP3 and pMLKL staining were reduced in the GLM group (Fig 5A and 5B). We next stimulated human OA chondrocytes with IL-1β, with or without GLM, and found that the transcriptional levels of RIPK1, RIPK3, and MLKL were substantially suppressed by GLME, consistent with the immunohistochemical data (Fig 5C).

## Discussion

Of the various dietary supplements, GLM is favored by patients with OA. Several clinical studies have reported good results [26]. We found that GLM reduced the clinical and histological

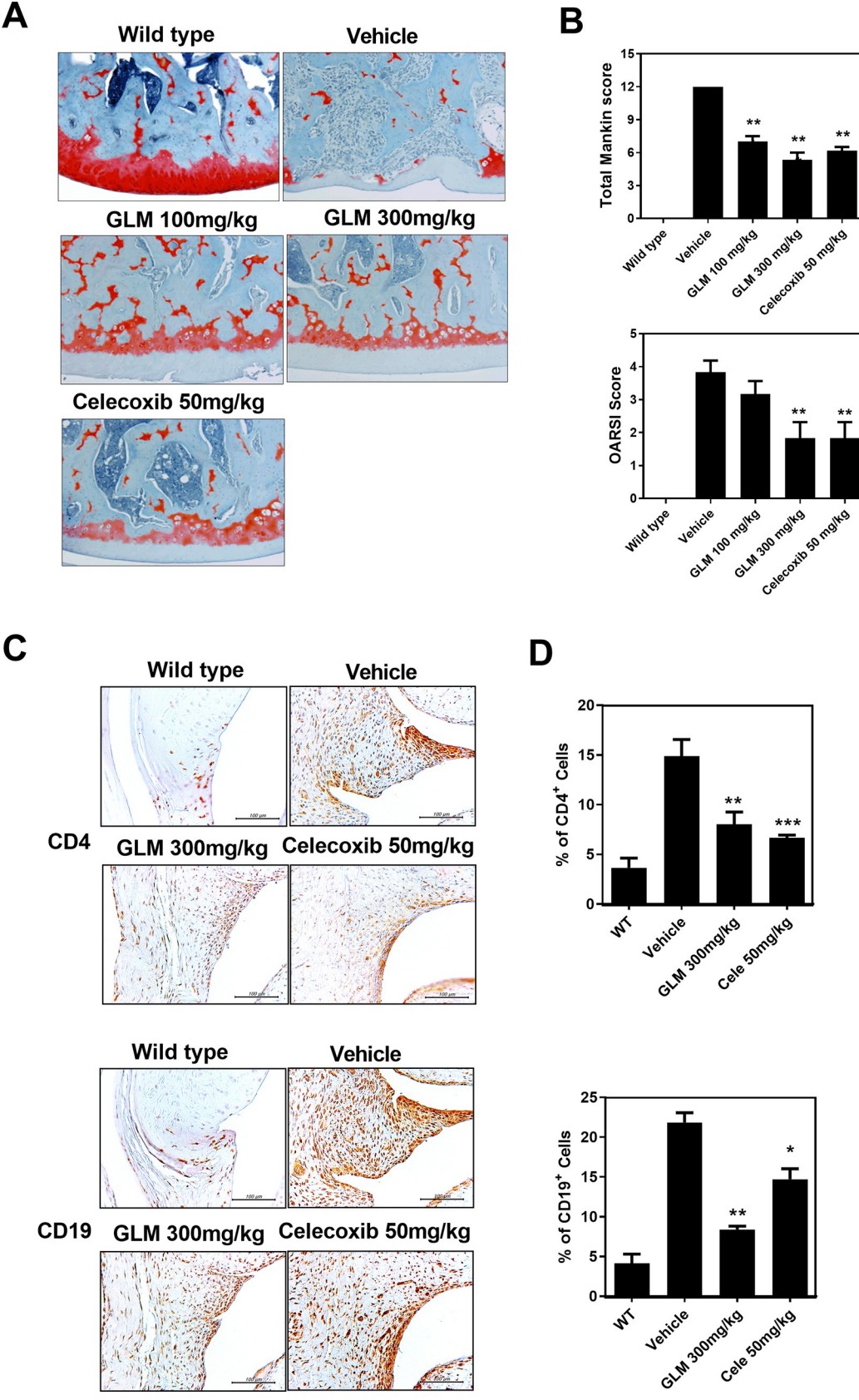

**Fig 2. Histological evaluation of joints after oral administration of GLME to rats with MIA-induced OA.** Rats were injected with 3 mg of monosodium iodoacetate (MIA) (into the right knee). GLME was given orally every day from day 3 after MIA injection. The joints were resected on day 21 after MIA injection. (A) Knee joints from the OA rats. Joint samples were acquired from the WT, vehicle, GLM (100 mg/kg, 300 mg/kg) and Celecoxib (50 mg/kg) groups and stained with hematoxylin and eosin (original magnification x 200). (B) The OA lesions were graded on a scale of 0–13 using the modified Mankin scoring system that evaluates structure, cellular abnormalities, and matrix staining. $^{*}P < 0.05$, $^{**}P < 0.01$, and $^{***}P < 0.001$ compared to the MIA-injection group. (C) CD4 and CD19 expression levels in the synovia of OA rats as revealed immunohistochemically 21 days after MIA injection. Immunohistochemistry was used to evaluate representative sections of joints from MIA-injected rats given GLM, celecoxib, or the vehicle. Positive cells stain brown; the nuclei were counterstained with hematoxylin. The bar graphs represent the means ± SDs of the numbers of stained cells. $^{*}P < 0.05$, $^{**}P < 0.01$ compared to the MIA-injection group.

scores of rats with MIA-induced OA. GLM significantly reduced pain, inflammatory cell infiltration, and the adverse histopathological changes. Thus, GLM simultaneously controlled OA pain and protected cartilage.

Articular cartilage degeneration is a major clinical feature of OA [27]. OA is not entirely a degenerative disease, being partly an inflammatory condition involving the actions of IL-1β, IL-6, and nitrotyrosine [28]. We found that GLM reduced cartilage injury and pain in the subchondral bone and the articular cartilage. IL-1β and IL-6 expression decreased significantly after GLM administration. The expression of iNOS was inhibited after GLME administration. The iNOS expression level reflects the NO content of OA joints. The role of NO in OA pathogenesis has recently been reported in detail [29]. In OA patients, control of oxidative stress is of therapeutic importance. This suggests that the reduced cartilage degradation in response to GLM was, in part, attributable to a lessening of oxidative damage and attenuated production of proinflammatory cytokines.

Persistent low-rate oxidative stress is a fundamental feature of OA pathology [30]. Increases in age-related oxidative stress render human chondrocytes susceptible to oxidant-mediated apoptosis via dysregulation of the antioxidant system [31]. Other forms of oxidative stress boost NO production, inducing aberrant apoptosis of human chondrocytes mediated via mitochondrial dysfunction [32]. MIA, a blocker of glyceraldehyde-3-phosphate dehydrogenase, generates OA-like lesions in rat joints because of cartilage degradation resembling that of human OA [33,34]. Turning to the significance of ROS and mitochondrial functional disorders in human OA, MIA-induced chondrocyte apoptosis has recently been reported to involve mitochondrial pathways including those responsible for ROS production and caspase activation [35]. These findings support the suggestion that the MIA model fairly reflects human OA; the cartilage breakdown mechanisms are similar. In our rat OA model, GLM reduced pain and eliminated the mitochondrial dysfunction caused by MIA.

Several recent reports have demonstrated the osteoarthritis therapeutic effectiveness of the administration of natural compound. Dietary supplementation with Palmitoyl-Glucosaine with curcumin decreased MIA-induced secondary allidynia at either threshold of latency level. Also, PGA-Cur with significantly decrease in serum levels of TNF-α, IL-1β, NGF as well as metalloproteases 1,3, and 9 [36]. In addition, there are studies on pain and inflammation ameliorate in osteoarthritis disease studies using natural compounds such as cashew nut, hyaluronic acid, and ALIAmide palmitoylglucosamine [21,22,37,38]. Also, in our studies GLM ameliorate pain and cartilage destruction in monosodium iodoacetate induced OA. The chondroprotective effects of GLM implicated inhibition of oxidative damage and suppression of catabolic factors involved in the pathogenesis of OA cartilage damage. But, further studies are need to further elucidate the relationship between natural compound and therapeutic effect of osteoarthritis.

The chondroprotective effects correlated strongly with catabolic inhibition of the articular cartilage matrix. The levels of specific markers of cartilage destruction, including MMP1,

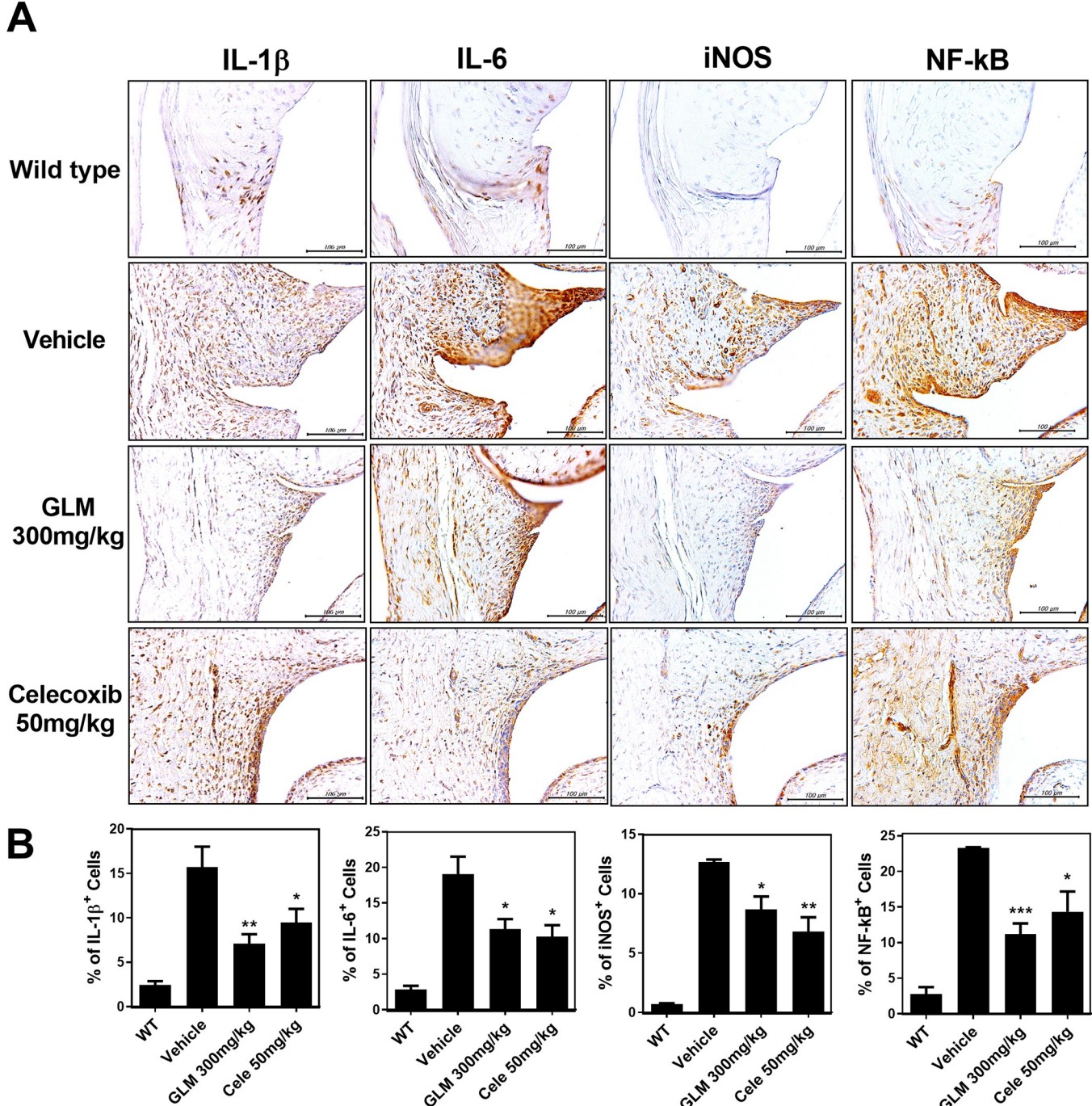

**Fig 3. Effects of GLME on the expression levels of IL-1β, IL-6, iNOS, and NF-kB in the synovia of OA joints.** (A) Immunohistochemical staining for IL-1β, IL-6, iNOS, and NF-kB after oral administration of GLME daily for 21 days after MIA injection. (B) The positive cells stain brown; the nuclei were counterstained with hematoxylin. Slides of representative sections showing expression of IL-1β, IL-6, iNOS, and NF-kB. $^{*}$P < 0.05, $^{**}$P < 0.01 compared to the MIA-injection group.

MMP3, and MMP13, were significantly decreased following GLM treatment of both rats and human OA chondrocytes. These valuable properties of GLM also delayed OA development, suggesting that the effects may be significant throughout the various stages of disease.

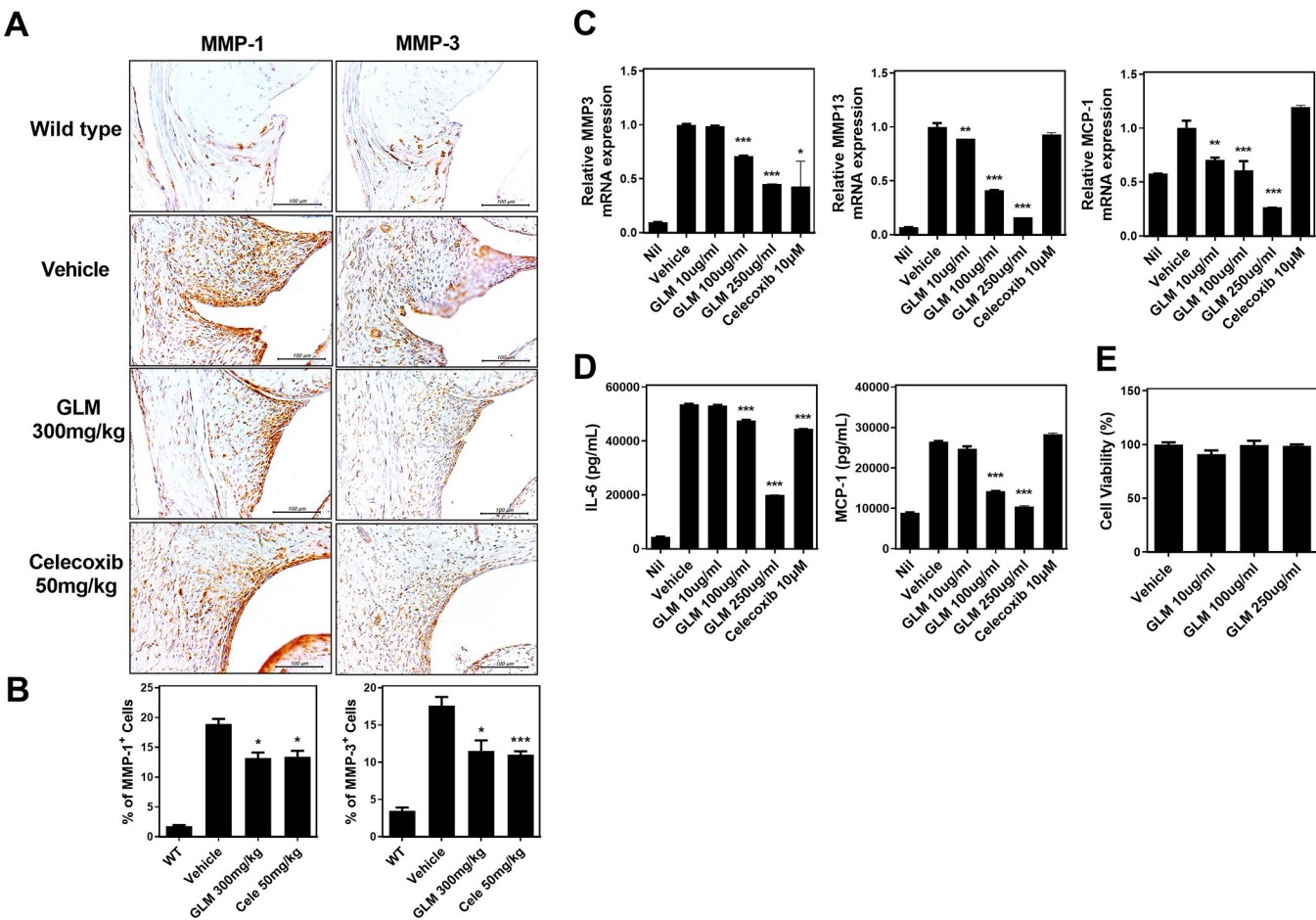

**Fig 4. Reductions in the levels of catabolic markers and inflammatory cytokines in OA rats and human chondrocytes.** (A, B) Immunohistochemical staining was used to identify MMP1 and MMP3. (C) qRT-PCR analysis of the levels of mRNAs encoding MMP3, MMP13 and MCP-1 in IL-1β (20 ng/mL)-stimulated OA chondrocytes in the presence or absence of (various concentrations of) GLME (10, 100, 250 μg/mL) and Celecoxib (10 μM). (D) IL-6 and MCP-1 levels in cell supernatants were measured via sandwich ELISAs. (E) OA chondrocyte cell viability in the presence or absence of GLME, as revealed by the CCK8 assay. *P < 0.05, **P < 0.01, and ***P < 0.001 compared to the vehicle-treated group.

Many studies have explored whether necroptosis inhibitors suppress cell death [39]. Necroptosis is significant in terms of inflammatory disease development. To the best of our knowledge, this is the first study to investigate the anti-necroptotic properties of GLME using both an animal model and human OA chondrocytes. GLM was decrease the levels of necroptotic mediators in the synovium of rats. GLM may attenuate the inflammatory response by inhibiting necroptosis.

GLME reduced NF-κB expression in rat joints. GLM controlled the levels of upstream mediators that modulate the expression of inflammatory cytokines. Thus, the anti-inflammatory effects of GLM may reflect reduced NF-κB activation. Further molecular characterization of the relevant factors is required.

In conclusion, To the best of our knowledge, this study is the first to demonstrate the potential of GLM as an OA therapeutic agent. MIA-induced arthritis was decreased by GLM, which inhibited catabolism, and the expression of proinflammatory cytokines and mediators of necroptosis. GLM may usefully prevent OA. Our data will aid OA patients who seek valuable natural products. GLM may be a useful OA therapeutic.

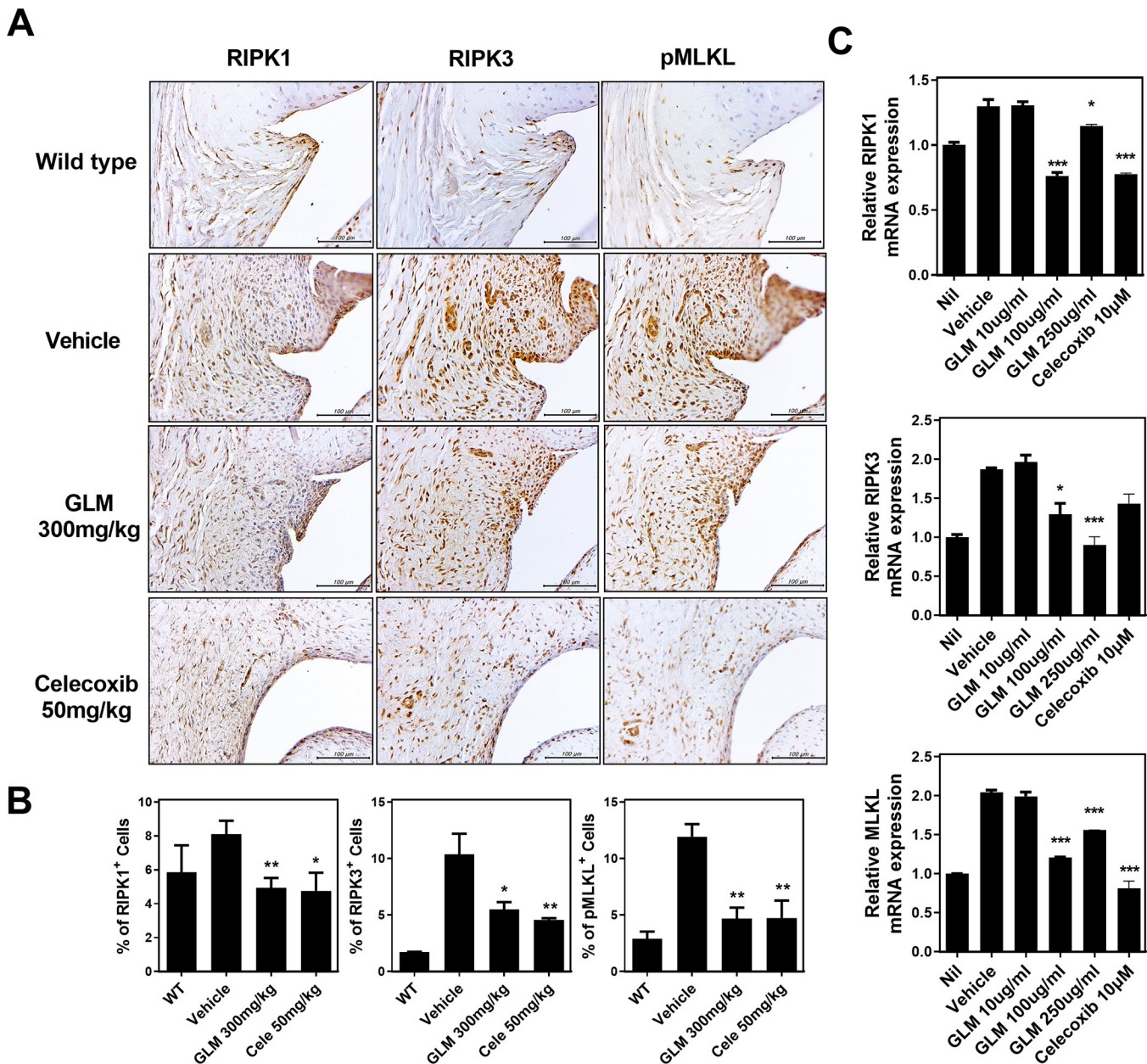

**Fig 5. Reduced levels of necroptosis-related markers in OA rats and human chondrocytes.** (A, B) Representative immunohistochemical staining of RIP1, RIP3, and pMLKL in the synovia of non-OA rats (WT), vehicle-treated MIA-induced OA rats, and GLME-treated MIA-induced OA rats. (C) The levels of mRNAs encoding necroptotic marker genes (as revealed by real-time PCR) in human OA chondrocytes treated with IL-1β (20 ng/mL) and then co-cultured with GLME or celecoxib. $^{*}P < 0.05$, $^{**}P < 0.01$, and $^{***}P < 0.001$ compared to the vehicle-treated group.

## Author Contributions

**Conceptualization:** JooYeon Jhun, Mi-La Cho.

**Data curation:** JooYeon Jhun, Hyun Sik Na, Keun-Hyung Cho, Jiyoung Kim, Young-Mee Moon, Seung Yoon Lee, A. Ram Lee.

**Formal analysis:** JooYeon Jhun, Hyun Sik Na, Jiyoung Kim, Jeong Su Lee, A. Ram Lee.

**Funding acquisition:** Sung-Hwan Park.

**Investigation:** JooYeon Jhun, Hyun Sik Na, Keun-Hyung Cho.

**Methodology:** Hyun Sik Na, Keun-Hyung Cho, Jiyoung Kim, Seung Yoon Lee, Jeong Su Lee, A. Ram Lee.

**Project administration:** JooYeon Jhun, Young-Mee Moon, Mi-La Cho.

**Resources:** Seok Jung Kim, Sung-Hwan Park.

**Supervision:** Seok Jung Kim, Mi-La Cho, Sung-Hwan Park.

**Validation:** JooYeon Jhun, Keun-Hyung Cho, Jiyoung Kim.

**Writing – original draft:** JooYeon Jhun, Hyun Sik Na.

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
