## [Decision Letter · Decision Letter 0]

7 May 2021

PONE-D-21-07398

A Green-lipped mussel reduces pain behavior and chondrocyte inflammation and attenuated experimental osteoarthritis progression

PLOS ONE

Dear Dr. 

Thank you for submitting your manuscript to PLOS ONE. After careful consideration, we feel that it has merit but does not fully meet PLOS ONE’s publication criteria as it currently stands. Therefore, we invite you to submit a revised version of the manuscript that addresses the points raised during the review process.

We look forward to receiving your revised manuscript.

Kind regards,

Rosanna Di Paola, MD

Academic Editor

PLOS ONE

Journal Requirements:

"NO - Include this sentence at the end of your statement: The funders had no role in study design, data collection and analysis, decision to publish, or preparation of the manuscript."

3. We noticed you have some minor occurrence of overlapping text with the following previous publication, which needs to be addressed:

- https://synapse.koreamed.org/articles/1033447

The text that needs to be addressed involves the Introduction section.

In your revision ensure you cite all your sources (including your own works), and quote or rephrase any duplicated text outside the methods section. Further consideration is dependent on these concerns being addressed.

Reviewers' comments:

Reviewer's Responses to Questions

**Comments to the Author**

1. Is the manuscript technically sound, and do the data support the conclusions?

Reviewer #1: Yes

Reviewer #2: Yes

2. Has the statistical analysis been performed appropriately and rigorously? 

Reviewer #1: Yes

Reviewer #2: I Don't Know

3. Have the authors made all data underlying the findings in their manuscript fully available?

Reviewer #1: Yes

Reviewer #2: Yes

4. Is the manuscript presented in an intelligible fashion and written in standard English?

Reviewer #1: Yes

Reviewer #2: No

5. Review Comments to the Author

Reviewer #1: The paper by JooYeon Jhun et al, describes the anti-inflammatory and joint-protecting properties of green-lipped mussel in a in vivo model of MIA.

The rational behind the experiment was clear and straight forward. The manuscript is almost well written

I recommend that the paper be accepted with minor revision:

a) The authors should better check the manuscript for any typographical errors.

b) The authors should clarify in the Materials and methods section the experimental groups and the experimental conditions.

c) The discussion does a good job at explaining the importance of the results in the context of the inflammatory pathways involved. However, incorporation of previous results from other related studies with natural compound is lacking such as 10.3390/ani10101827 or 10.3390/antiox9060511 or 10.1186/s13075-016-1189-5 or 10.1186/s13075-019-2048-y or 10.1155/2021/5534614 or 10.3390/antiox10020265

d) please provide high magnification of figure 2,3,4 and 5 to better understand the signal maybe as supplementary files and better clarify the method for IHC quantification

Reviewer #2: The manuscript by JooYeon Jhun al ” A Green-lipped mussel reduces pain behavior and chondrocyte

inflammation and attenuated experimental osteoarthritis progressionnts”

The manuscript is interesting in its field, showing an overview of the GLM use in osteoarthritis progressionnts.

I recommend major revision for the paper:

• The authors should improve the english language

• The author should update the bibliography. For example in the introduction the author should insert several paper about the anti inflammatory and antioxidative application of compound present in diet(doi.org/10.3390/antiox9060511;oi.org/10.1186/s13075-019-2048-y)

• the authors should indicate the experimental design through a temporal sequence, clearly indicating the administrations and tests carried out

• The authors should increase the quality of the images, for example in figure 2C, the photos appear to have been taken in different places

• Is there a mechanism of action for GLM? The authors speak of a mediated NfkB mechanism, has a mechanism directly linked to GLM been hypothesized?

6. PLOS authors have the option to publish the peer review history of their article (what does this mean?). If published, this will include your full peer review and any attached files.

Reviewer #1: No

Reviewer #2: No

---

## [Author Response · Author response to Decision Letter 0]

5 Oct 2021

Reviewers' comments:

Reviewer's Responses to Questions

Comments to the Author

1. Is the manuscript technically sound, and do the data support the conclusions?

Reviewer #1: Yes

Reviewer #2: Yes

2. Has the statistical analysis been performed appropriately and rigorously?

Reviewer #1: Yes

Reviewer #2: I Don't Know

3. Have the authors made all data underlying the findings in their manuscript fully available?

The PLOS Data policy requires authors to make all data underlying the findings described in their manuscript fully available without restriction, with rare exception (please refer to the Data Availability Statement in the manuscript PDF file). The data should be provided as part of the manuscript or its supporting information, or deposited to a public repository. For example, in addition to summary statistics, the data points behind means, medians and variance measures should be available. If there are restrictions on publicly sharing data?e.g. participant privacy or use of data from a third party?those must be specified.

Reviewer #1: Yes

Reviewer #2: Yes

4. Is the manuscript presented in an intelligible fashion and written in standard English?

Reviewer #1: Yes

Reviewer #2: No

5. Review Comments to the Author

Reviewer #1: The paper by JooYeon Jhun et al, describes the anti-inflammatory and joint-protecting properties of green-lipped mussel in a in vivo model of MIA.

The rational behind the experiment was clear and straight forward. The manuscript is almost well written

I recommend that the paper be accepted with minor revision:

a) The authors should better check the manuscript for any typographical errors.

Answers: Thank you for your comments. As you mentioned that we checked error and typographical. 

b) The authors should clarify in the Materials and methods section the experimental groups and the experimental conditions.

Answers: Thank you for your comments. Osteoarthritis was induced in Wistar rat (n=6). The experiment was performed two times. We revised the Material and method section (marker by red color, Page 6, line 21-23) 

The rats were randomly assigned to treatment groups prior to the beginning of the study. Rats were divided into six groups (5 rats in each group) and were group housed in two cages. Each in vivo experiment was repeated a total 2 times.

c) The discussion does a good job at explaining the importance of the results in the context of the inflammatory pathways involved. However, incorporation of previous results from other related studies with natural compound is lacking such as 10.3390/ani10101827 or 10.3390/antiox9060511 or 10.1186/s13075-016-1189-5 or 10.1186/s13075-019-2048-y or 10.1155/2021/5534614 or 10.3390/antiox10020265

Answers: Thank you for valuable suggestion. This comment added the discussion section. (marker by red color, Page 14, line 4-15)

Several recent reports have demonstrated the osteoarthritis therapeutic effectiveness of the administration of natural compound. Dietary supplementation with Palmitoyl-Glucosaine with curcumin decreased MIA-induced secondary allidynia at either threshold of latency level. Also, PGA-Cur with significantly decrease in serum levels of TNF-�, IL-1�, NGF as well as metalloproteases 1,3, and 9 [36]. In addition, there are studies on pain and inflammation ameliorate in osteoarthritis disease studies using natural compounds such as cashew nut, hyaluronic acid, and ALIAmide palmitoylglucosamine [21, 22, 37, 38]. Also, in our studies GLM ameliorate pain and cartilage destruction in monosodium iodoacetate induced OA. The chondroprotective effects of GLM implicated inhibition of oxidative damage and suppression of catabolic factors involved in the pathogenesis of OA cartilage damage. But, further studies are need to further elucidate the relationship between natural compound and therapeutic effect of osteoarthritis. 

d) please provide high magnification of figure 2,3,4 and 5 to better understand the signal maybe as supplementary files and better clarify the method for IHC quantification

Answers: Thank you for your comment. All images have been replaced with high magnification. Immunohistochemistry evaluations were performed independently by two experienced researchers who were blinded to the study groups. The positive cell percentage was analyzed with HDAB (Hematoxylin & DAB) by selecting color deconvolution in the plugin item in the image J program (NIH, MD ,USA). We revised the Material and method section (marker by red color, Page 8, line 15 -18 )

Reviewer #2: The manuscript by JooYeon Jhun al " A Green-lipped mussel reduces pain behavior and chondrocyte

inflammation and attenuated experimental osteoarthritis progressionnts"

The manuscript is interesting in its field, showing an overview of the GLM use in osteoarthritis progressionnts.

I recommend major revision for the paper:

? The authors should improve the english language

Answers: Thank you for your comment. The English in this document has been checked by at least two professional editors, both native speakers of English. For a certificate, please see:

http://www.textcheck.com/certificate/qUhLlk

? The author should update the bibliography. For example in the introduction the author should insert several paper about the anti-inflammatory and antioxidative application of compound present in diet(doi.org/10.3390/antiox9060511;oi.org/10.1186/s13075-019-2048-y)

Answers: Thank you for your comment. Thanks for suggesting a great article and we’ve added the contents of the article on introduction. (marked by red color, page 6, line 12-14), (marked by red color, page 4, line 5-10) 

Also, previous study showed that osteoarthritis development is notably associated with oxidative stress and ROS [19, 20]. There are many studies of its application in the oral administration of osteoarthritis through anti-inflammatory and antioxidant effects [21, 22]. In particular, GLM have antioxidant activity. Bioactive peptides of GLM with the strongest radical scavenging activity and ACE inhibitory activity would supply a therapeutic effect for the osteoarthritis [23].

? the authors should indicate the experimental design through a temporal sequence, clearly indicating the administrations and tests carried out

Answers: Thanks for your comment. We agree with your comments. This comment added the Figure 1.

The protocol for monosodium iodoacetate induced osteoarthritis and ? The authors should increase the quality of the images, for example in figure 2C, the photos appear to have been taken in different places

Answers: Thank for your comments. We agree with your opinions. All images have been replaced with high magnification. As you mentioned, these images are synovium, not a cartilage. Imaging synovium makes it difficult to identify differences in immunohistochemistry in cartilage because the Monosodium Iodoacetate Arthritis model affects cartilage destruction and chondrocyte death. Also, synovitis symptoms appeared due to the characteristics of the MIA models, so Immunohistochemistry was performed in synovium. The quality of the image has been increased. In addition, a separate enlarged shot is included in the image so that you can see it clearly. 

? Is there a mechanism of action for GLM? The authors speak of a mediated NfkB mechanism, has a mechanism directly linked to GLM been hypothesized?

Answers: Thanks for pointing out this and we’ve added other taxa following your suggestion. (marked by red color, page 4, line 11-13)

And previously studies have reported the nuclear factor-kappaB (NF-�B) transcription factors as over activated in OA and as a diseases-providing factor [7, 8]

6. PLOS authors have the option to publish the peer review history of their article (what does this mean?). If published, this will include your full peer review and any attached files.

Do you want your identity to be public for this peer review? For information about this choice, including consent withdrawal, please see our Privacy Policy.

Reviewer #1: No

Reviewer #2: No

---

## [Editor Report · Decision Letter 1]

14 Oct 2021

A Green-lipped mussel reduces pain behavior and chondrocyte inflammation and attenuated experimental osteoarthritis progression

PONE-D-21-07398R1

Dear Dr.

We’re pleased to inform you that your manuscript has been judged scientifically suitable for publication and will be formally accepted for publication once it meets all outstanding technical requirements.

Kind regards,

Rosanna Di Paola, MD

Academic Editor

PLOS ONE
---

## [Editor Report · Acceptance letter]

23 Nov 2021

PONE-D-21-07398R1 

A Green-lipped mussel reduces pain behavior and chondrocyte inflammation and attenuated experimental osteoarthritis progression. 

Dear Dr. Cho:

I'm pleased to inform you that your manuscript has been deemed suitable for publication in PLOS ONE. Congratulations! Your manuscript is now with our production department. 

Kind regards, 

on behalf of

Dr. Rosanna Di Paola 

Academic Editor

PLOS ONE